# Shared-Weight-Based Multi-Dimensional Feature Alignment Network for Oriented Object Detection in Remote Sensing Imagery

**DOI:** 10.3390/s23010207

**Published:** 2022-12-25

**Authors:** Xinxin Hu, Changming Zhu

**Affiliations:** College of Information and Engineering, Shanghai Maritime University, Shanghai 201306, China

**Keywords:** remote-sensing images, oriented object detection, feature alignment, anchor free, convolutional neural networks

## Abstract

Arbitrarily Oriented Object Detection in aerial images is a highly challenging task in computer vision. The mainstream methods are based on the feature pyramid, while for remote-sensing targets, the misalignment of multi-scale features is always a thorny problem. In this article, we address the feature misalignment problem of oriented object detection from three dimensions: spatial, axial, and semantic. First, for the spatial misalignment problem, we design an intra-level alignment network based on leading features that can synchronize the location information of different pyramid features by sparse sampling. For multi-oriented aerial targets, we propose an axially aware convolution to solve the mismatch between the traditional sampling method and the orientation of instances. With the proposed collaborative optimization strategy based on shared weights, the above two modules can achieve coarse-to-fine feature alignment in spatial and axial dimensions. Last but not least, we propose a hierarchical-wise semantic alignment network to address the semantic gap between pyramid features that can cope with remote-sensing targets at varying scales by endowing the feature map with global semantic perception across pyramid levels. Extensive experiments on several challenging aerial benchmarks show state-of-the-art accuracy and appreciable inference speed. Specifically, we achieve a mean Average Precision (mAP) of 78.11% on DOTA, 90.10% on HRSC2016, and 90.29% on UCAS-AOD.

## 1. Introduction

Benefiting from remote-sensing images with a broad field of view and high image quality, object detection has been widely used in various civilian and military fields, such as inshore vessel monitoring, geological exploration, and traffic diversion. With many convolutional neural networks (CNNs) and transformer-based detection algorithms proposed in the natural image domain [1,2,3,4,5,6,7], the performance of horizontal object detection has been significantly improved. However, the horizontal bounding boxes used for natural images are unsuitable for remote-sensing targets with dense arrangements and arbitrary orientations. In this case, an oriented bounding box (OBB) is a more domain-friendly solution that can perfectly distinguish between instances at high camera altitudes while reducing the redundant background inside the boxes.

Still, this annotation approach also poses several domain-specific challenges. For small remote-sensing targets with variable orientations in dense scenes, high localization accuracy and orientation consistency of detection boxes are required. For remote-sensing images with severe foreground–background imbalance and complex contexts, robust semantic representation of inter- and intra-class features is demanded. In this work, we summarize the misalignment problems that exist in oriented object detection from three aspects:(i)**Axial misalignment**: The feature extraction method of traditional convolution filters the whole image in the form of a sliding window under a fixed coordinate system, and the sampling axis of the kernel is consistent with the sliding direction. However, instance details are lost to a certain extent due to high-altitude shooting in remote-sensing imagery, which makes different instances of the same category have similar textures, while these similar targets belong to different axes in the bird’s eye view, as shown in Figure 1a. CNNs have weak axial perception, and even with the help of resampling operations such as pooling, they are still not robust enough. For a similar object in a different axis, the CNN tends to learn it as a new object instead of the rotated one [8]. Therefore, this fixed axial sampling method does not match the orientation of remote-sensing targets and OBBs. Eventually, the axial misalignment problem makes it difficult for the extracted features to represent arbitrarily oriented targets well, resulting in inaccurate regression, as shown in Figure 1b.(ii)**Spatial misalignment**: First, in the process of extracting multi-scale features from the backbone, the aliasing effect caused by the interpolation, the weak response loss caused by pooling, and the channel compression all cause different degrees of location information loss. Secondly, remote-sensing targets are small and densely arranged and vary widely in shape, rotation, and position, so this information loss significantly impacts the remote-sensing field more than other fields. The spatial misalignment problem in the detection stage leads to inaccurate regression, which further causes the predicted boxes to be incorrectly filtered by post-processing operations, as shown in Figure 1c.(iii)**Semantic misalignment**: Since the semantic information in the backbone features is distributed in several pyramid levels due to the receptive field differences, there is a huge semantic gap between layers. Although a feature pyramid network (FPN) [9] alleviates the above problem by a top-down feature fusion, its linear combination of the feature hierarchy cannot perfectly cope with the large inter- and intra-class scale variations of remote-sensing targets. Therefore, classification errors due to semantic non-alignment between pyramid levels still exist, as shown in Figure 1d.

For the above misalignment problems, many excellent algorithms have achieved considerable performance and can be divided into three directions. The first is the sampling method based on predicted boxes: [10,11,12] refine the feature points under the guidance of the key points on the rotated anchors. However, these methods rely on an additional refine head, which greatly impacts inference efficiency and parameters. Another direction is the fusion method based on feature resampling: [13,14,15,16,17,18] aggregate features from all pyramid levels and finally predict at the specified feature scale. The disadvantage of these methods is that resampling features to a single scale weakens the scale sensitivity of the network.

In this paper, we propose an intra-level alignment network based on sparse sampling and axially aware convolution for the misalignment problem in axial and spatial dimensions. The network jointly absorbs the location information and semantic knowledge from low-level and high-level features, respectively, to achieve a coarse-to-fine form of feature alignment. In addition, a collaborative optimization strategy based on shared weights is proposed to make feature alignment more stable and robust. Next, we propose a hierarchical-wise context fusion modulethat enhances the global semantics perceptivity of each level while dynamically fusing the pyramid features to achieve semantic feature alignment between different levels. Finally, our proposed method achieves remarkable performance on several public remote-sensing datasets (e.g., DOTA1.0, HRSC2016, and UCAS-AOD) and achieves accuracy comparable to state-of-the-art methods.

The key contributions of this paper are as follows:First, we design a shared-weight-based multi-scale alignment network to synchronize the spatial information of different pyramid features. Following this, an axially aware convolution is proposed to align the sampling axis with remote-sensing targets in arbitrary orientations. So far, we have achieved the feature alignment of spatial and axial dimensions in a coarse-to-fine manner without the help of additional detection heads.Secondly, a hierarchical-wise context fusion module is proposed to cope with the domain-specific problem of large intra-class scale variation by aligning semantic knowledge among pyramid features and fusing them dynamically.Empirically, we demonstrate the effectiveness of the proposed methods through qualitative and quantitative ablation experiments. Comparing baseline and state-of-the-art methods, our OA-Det achieves comparable performance on three challenging datasets.

The rest of the paper is organized as follows. Section 2 begins with a brief introduction of the work related to this paper from three aspects. Then, the proposed feature alignment method is introduced from the three dimensions of space, axis, and semantics. Section 3 introduces the datasets and implementation details used for the experiments and then analyzes the ablation experiments of each module and the main results on the test set. Section 4 objectively discusses the limitations and future direction of the proposed method. Conclusions are given in Section 5.

## 2. Materials and Methods

### 2.1. Related Work

In this chapter, we focus on the domain-related work involved in the proposed methods. First, we introduce horizontal box-based object detection in terms of the execution stage of the algorithm and the detector type. Next, we list five representative algorithms for arbitrarily oriented object detection according to the type of method. Finally, we introduce the existing solutions related to feature alignment from both spatial and semantic perspectives, and the differences between our methods and existing methods are analyzed.

#### 2.1.1. Horizontal Object Detection

Object detection based on horizontal bounding boxes has seen a remarkable leap in the last decade. From the traditional method based on handcrafted features [19,20], to the near-invincible deep convolutional neural network [21], to the Transformer architecture [22] that has emerged in recent years, we have been provided with the foundational support for a variety of visual detection algorithms. According to the execution stage of the algorithm, they can be divided into two-stage detectors, represented by [2,4], and single-stage detectors, represented by [6,7,23]. With its concise end-to-end network structure, the single-stage detector achieves a good trade-off in speed, parameters, and accuracy.

In recent years, anchor-free single-stage detectors have been widely proposed. CenterNet [24] predicts a center point for each object, regressing the width and height of the horizontal bounding box directly from the center point. RepPoints [25] characterize the target by adaptively learning a set of representative sampling points. FCOS [26] regresses the distances from the predicted points to the four boundaries to obtain the horizontal prediction box.

Compared with the anchor-based method, using the point as the starting state of the regression can better cope with pixel-level dense scenes. Further, the anchor-free method without anchor box hyperparameters has a more efficient inference speed, so the architecture used in this paper is a center-based anchor-free detector.

#### 2.1.2. Arbitrarily Oriented Object Detection in Aerial Images

Remote-sensing images have more complex contextual relationships between background and foreground, and the targets to be detected have the characteristics of multi-oriented, complex backgrounds, dense arrangements, and extreme scale differences. Therefore, oriented object detection needs to address plenty of domain-specific challenges. SCRDet [27] proposes a feature fusion network based on saliency maps from the perspective of an attention mechanism to cope with small and dense remote-sensing targets. CFC-Net [28] proposes a task-specific polarization function to solve the feature incompatibility problem of classification and regression tasks; ReDet [8] extracts rotation-invariant features by reconstructing the backbone network so that the feature maps are discriminative for objects with arbitrary orientations. SASM [29] proposes a shape-sensitive label assignment method that is suitable for high aspect ratio remote-sensing targets. RBox [30] improves the performance of the Transformer architecture in rotated object detection and solves the problem of redundant feature backgrounds.

Our method is dedicated to solving feature misalignment problems for small and multi-directional remote-sensing targets and is applied to three dimensions of spatial, axial, and semantic.

#### 2.1.3. Feature Alignment in Object Detection

The problem of feature non-alignment generally exists in feature pyramid networks. Related research on feature alignment can be divided into two main research lines:

One line of research focuses on feature alignment in the spatial dimension. R${}^{3}$Det [13] uses the best prediction box to guide feature interpolation in the refined stage. The authors of [31] predict 2D offsets for each position on the feature map in semantic segmentation tasks. These methods directly perform spatial reconstruction of feature maps in a lightweight manner. However, it is difficult to guarantee alignment stability only by relying on the predicted 2D offset map. After that, a great deal of work attempted to guide sampling locations. FaPN [32,33] combines shallow and deep features to guide the location of each sampling point; [12,34,35,36,37,38] adapt the sampling points to high-aspect-ratio remote-sensing targets through the rotated bounding box predicted by the additional head.

Another line of research focuses on feature alignment in the semantic dimension. CGNet [39] uses self-attention to enhance communication between pyramid levels. AFF-Det [14,18] maps ROIs to all levels and applies a unified supervisory signal to alleviate the semantic gap. The authors of [16,17] perform feature refinement after scaling. SCRDet++ [40] introduces instance-level denoising to decouple intra-class and inter-class features.

The above methods solve domain-related problems, but the extra detection heads increase the cost of parameters and computation. On the other hand, recursive resampling introduces cross-layer semantic interference. Our intra-level collaborative feature alignment network can act in axial and spatial dimensions without extra supervision signals. In addition, the proposed semantic alignment module addresses the semantic gap in multi-scale features.

### 2.2. The Proposed Method

#### 2.2.1. Overall Pipeline

The proposed OA-Det is built based on the basic center-based anchor-free detector. The overall architecture consists of four components: the feature extraction backbone, the axially aware spatial alignment network, the hierarchical-wise semantic alignment network, and the oriented detection heads, as shown in Figure 2.

Specifically, we first feed an RGB remote-sensing image as input to the feature extraction backbone network. We adopt ResNet101 [41] pretrained on ImageNet [42] as the backbone network to generate four different scale feature maps$\left\{C_{2},C_{3},C_{4},C_{5}\right\}$, where $C_{i}$ has $\frac{1}{2^{i}}$ resolution of the input image. Next, we construct a feature pyramid network to transfer the semantic information of the deep network to the shallow features layer-by-layer. We first downsample $C_{5}$ using 3×3 convolution to obtain features with smaller resolution and richer semantic information for adapting to larger-scale remote-sensing instances. Then, we fuse the pyramid features with ResNet features in a top-down manner using lateral connections and interpolation. So far, we can get the pyramid features $\left\{P_{2},P_{3},P_{4},P_{5},P_{6},P_{7}\right\}$. The detection algorithm based on the above multi-scale features can better cope with large-scale variations of remote-sensing objects. It should be noted that $C_{2}$ is also fused to obtain $P_{2}$ with richer location information. However, we only feed it to the spatial alignment network instead of the final detection head for prediction. More details can be found in Section 2.2.2. Then, the intra-level collaborative alignment network performs spatial information synchronization for each pyramid feature, followed by fine feature sampling using axially aware convolution. Based on the above aligned features, we use a hierarchical-wise context fusion module to enhance the global information from the channel dimension and perform dynamic feature fusion between pyramid levels. Finally, the multi-scale features aligned in spatial, axial, and semantic dimensions are sent to task-specific detection heads. We construct the classification head and regression head by several stacked convolutional layers to predict the probability of the category and the distance vector for each position *i* in the feature map, respectively. The distance vector can be formulated as follows:(1)$$V_{i}=\left(l_{i},t_{i},r_{i},b_{i}\right),$$
where $l_{i},t_{i},r_{i},b_{i}$ are the distances to the left, top, right, and bottom boundaries of the box; ${\mathrm{centerness}}_{i}$ [26] is predicted on another branch to suppress the low-quality bounding boxes that deviation from the center point of the ground truth. It can be calculated by the following equation:(2)$${\mathrm{centerness}}_{i}=\sqrt{\frac{min(l_{i},r_{i})}{max(l_{i},r_{i})}\times \frac{min(t_{i},b_{i})}{max(t_{i},b_{i})}}.$$

For arbitrarily oriented object detection, the rotated bounding boxes are usually expressed by $\left(x_{c},y_{c},h,w,\theta \right)$, where $x_{c}$, $y_{c}$ are the center point of the box, and h,w,θ represent the height, width, and angle of the rotated box, respectively. Thus, we add a parallel branch to the regression head to predict θ, where angle representation is referenced to [11]. Finally, the rotated bounding box can be obtained by combining the angle θ with the above distance vector $V_{i}$. Next, we present a detailed description of the proposed feature alignment module.

#### 2.2.2. Shared-Weight-Based Multi-Scale Feature Alignment Network Using Axial-Aware Convolution

The problem of spatial misalignment is a long-standing problem for multi-scale features. Due to the recursive use of resampling operations in the process of extracting multi-scale features, this problem is gradually exacerbated, and the impact on remote-sensing object detection is even more immeasurable. On the one hand, because of the small and dense arrangement of remote-sensing targets, as long as there is a slight deviation in the deep features, the predicted boxes fall to the adjacent instance. On the other hand, there is also a domain-specific axial misalignment problem in oriented object detection that is caused by the mismatch between the sampling method of regular convolution and the orientation of remote-sensing targets.

In this section, we first design an intra-level collaborative feature alignment network to deal with the spatial misalignment problem. After that, an axially aware convolution is proposed to alleviate the axial misalignment problem of sampling points. The architecture is shown in Figure 3. First, we take the pyramid feature $P_{n}$ as the leading feature and perform adaptive spatially aligned resampling of features at adjacent levels; the implementation detail is as follows:

In order to obtain the prior position information of the feature maps, we predict *K* sets of 2D offset maps from the leading feature, where *K* is the number of sampling points at each position. The expression is as follows:(3)$$\begin{array}{cc} \; & \Delta p^{\left(n\right)}=f_{offset}\left(P_{n}\right), \end{array}$$
where $f_{offset}\left(\cdot \right)$ is a convolution layer and $\Delta p^{\left(n\right)}$ is the offset map predicted from the *n*-th level of pyramidal features. Each offset value consists of a tuple $\left(x_{k},y_{k}\right)$ that is used to guide each sampling point. Next, we implement spatial alignment for neighbor features:(4)$${\hat{P}}_{\ell }=A\left(∥P_{\ell }∥,\Delta p^{\left(n\right)}\right),$$
where ∥·∥ represents the scaling operation and ${\hat{P}}_{\ell }$ represents the feature map with pyramid level *ℓ* after spatial alignment. The sparse sampling method A(·) is implemented using deformable convolution [43]. Note that during the sampling process, we share offset maps for neighbor features to ensure the consistency of position information within the pyramid level. Next, we focus on how to calculate each point on the feature map, which can be expressed as:(5)$${\hat{x}}_{\ell }\left(p\right)=\sum_{k=1}^{K}w_{\ell }\left(p_{k}\right)\cdot x_{\ell }\left(p+p_{k}+\Delta p_{k}^{\left(n\right)}\left(p\right)\right),$$
where ${\hat{x}}_{\ell }\left(p\right)$ is the value of the spatially aligned feature map at position *p*; $w_{\ell }\left(\cdot \right)$ and $x_{\ell }\left(\cdot \right)$ are kernel weight and input feature value, respectively; and $p_{k}$ is the sampling grid. When the convolution kernel is 3 and the dilation rate is 1, $p_{k}\in \left\{(-1,-1),(-1,0),\ldots ,(1,1)\right\}$.

So far, we have obtained multiple neighbor features at each pyramid level with resolution and spatial information consistent with the leading features. Compared with the non-learnable resampling in FPN, sparse sampling based on shared offsets can reconcile the spatial information of neighbor features and leading features, thereby solving the problem of spatial misalignment.

Next, for the domain-specific axial non-alignment problem, the most commonly used method is alignment convolution [11]. The core idea is to use refined heads to make a pre-prediction before the downstream task and then use the predicted rotated box to guide the sampling position of the convolution. After our observations, this method has the following three shortcomings:(i)First, this anchor-aware sampling method tends to introduce foreign background interference. Since the offset is calculated from the width and height of the predicted boxes, although high-aspect-ratios targets are well adapted, there is still a lot of background inside the rotated boxes for targets with irregular shapes, as shown in Figure 4a.(ii)The representation of the sampling axis in alignment convolution is unique. Figure 4b shows the axial representation of different sampling methods. The alignment convolution predicts only one box at each position, and each positive sample is learned from a single ground truth. Hence, the sampling axis of each convolution kernel is in the same orientation as the rotated box, for which the representation is unique. However, the sampling axis of the convolution kernel is not affected by the angular periodicity problem [44,45]. Given the same sampling shape, an axis can be represented by four angles to achieve the same sampling effect.(iii)Additional supervision signals and anchor-based sub-heads increase the number of parameters and amount of computation.

Through the above analysis, we propose an axially aware convolution to deal with the domain-specific axial misalignment problem. The overall process of axial alignment is as follows: To accurately perceive the target’s orientation, we need to make full use of the neighbor features that have been aligned in spatial dimension. So we first fuse these features by element-wise addition and feed them into the spatially aware block, which can be formulated as:(6)$$\begin{array}{cc} \; & X_{\mathrm{sa}}={\hat{P}}_{n-1}+{\hat{P}}_{n}+{\hat{P}}_{n+1}\\ \; & {\hat{X}}_{\mathrm{sa}}=\mathrm{Proj}\left(F_{c}\left(F_{s}\left(X_{\mathrm{sa}}\right)\right)\right)+X_{\mathrm{sa}}, \end{array}$$
where $F_{s}$ and $F_{c}$ represent the feature aggregation in spatial and channel dimensions, respectively. We use 7 × 7 depth-wise convolution combined with point-wise convolution to implement the above operation. Our idea is to extract local spatial information with a large receptive field using the group convolution of the big kernel. At the same time, we use two successive 1×1 convolutions to achieve multi-layer perception(MLP)-like channel projections, which can enhance the generalizability of features at a low cost.

So far, the aggregated features contain rich semantic knowledge and localization information from the upper and lower levels, respectively. We then apply the proposed axially aware convolution for refined sampling. Different from directly predicting the offset tuple, we predict a θ at each position to characterize the orientation of the sampling point set. In addition, an activation scalar is predicted to indicate the importance of each sample point after shifting [46]. The expression is as follows:(7)Δθ(n)=faxis(X^sa),Δw(n)=fmask(X^sa),
where faxis and fmask are 3 × 3 convolution. We predict an axis map Δθ(n) and an L scalar map Δw(n) for each pyramid level. It is worth noting that we use the same weights between layers to predict the axis map, while for the scalar map, we also share weights within each level. In this way, we endow the maps with the same implicit supervision signal to achieve cross-layer axial alignment. Eventually, the finely aligned feature map can be obtained by axially aware sampling:(8)y^ℓ(p)=∑k=1Kwℓpk·x^ℓp+pk+Δℙk(n)(p)·Δwk(n)(p),Δℙk(n)=M(Δθ(n))·pk−pk,
where Δℙk(n) are the axially aware offsets for each sampling point, M(·) represents an affine rotation matrix. In contrast to Equation (5), the first difference is that the offset of the K sampling points is not directly predicted but is calculated by an axis shared across levels. Thus, the convolutional kernel is endowed with axial perception ability. Secondly, each offset sample point is re-weighted to avoid interference from redundant backgrounds. In this work, we do not compute the loss of θ with the ground truth in an explicitly supervised manner. The reason is that we consider predicting the regression box and guiding the sampling points as different tasks. The former needs to accurately predict the angle of the rotation box, which is unique, while the axial representation can be various. Therefore, additional supervision will limit the generalization ability of axial perception. Comparative experiments of different axial alignment methods can be found in Section 3.3.3. Further, in order to ensure the stability of the predicted axis, we assign the shared weights to collaboratively optimize the axis at each position. See Section 3.3.4 for ablation experiments on weight strategies.

Correctly choosing the pyramid level that needs to be aligned is also a point that cannot be ignored. First, the location-rich $P_{2}$ is introduced to guide the upper-level features for more accurate spatial and axial alignment. However, we do not feed it into the detection head for final prediction. In this way, we can cope with dense predictions effectively. Second, since the low-resolution features contain little spatial information, the network does not act on the high-level pyramid feature $P_{7}$. Nevertheless, our alignment networks have multiple in-degrees for each level of feature data flow during forward and backward propagation, so it can actually affect all pyramid features. Specifically, in the process of network forward inference, features can obtain rich semantic knowledge and location information from adjacent levels. Moreover, during backward propagation, the features can obtain the reverse gradient from the resampled features provided to other levels. More detailed alignment layer ablation experiments are shown in Section 3.3.2.

#### 2.2.3. Hierarchical-wise Context Fusion Module for Semantic Alignment

So far, we have obtained neighbor features aligned in spatial and axial dimensions within each pyramid level, but there is still a problem of semantic misalignment between them. When faced with remote-sensing targets with variable scales, even if the OBBs are correctly regressed, classification errors still exist due to the semantic differences between levels. In the case of a complex contextual relationship between the background and the foreground of aerial images, the foreground is overwhelmed by a large number of object-like backgrounds. The mainstream method [9] conveys the semantic information layer-by-layer using lateral connections. Other improved works [13,14,15,16,17,18] apply semantic reconfiguration to single-scale aggregated features resampled from all pyramid levels. However, we find that these methods have certain shortcomings:First, fusing all pyramid features with different receptive fields into a single scale introduces a lot of irrelevant semantic information. Figure 5a demonstrates the correlations of the backbone features and pyramid features. We can find that the values near the diagonal are significantly larger than the remote ones, whether before or after feature fusion, indicating that the features of each layer are only most correlated with neighbor features. Therefore, it is not appropriate to fuse all features after resizing them to the same scale.Secondly, feature fusion based on lateral connections is equivalent to a linear combination of the feature hierarchy, which is weak in capturing highly nonlinear features. Thus, it is difficult to cope with remote-sensing targets with various shapes and scales.These methods ignore feature misalignment in spatial and axial dimensions before fusing multi-scale features, resulting in semantic interference across levels.

In order to solve the above problems, we propose a hierarchical-wise global context fusion module. As shown in Figure 6, we take the spatially and axially aligned features as input and first perform global contextual semantic modeling for each pyramid level, which can be formulated as:(9)Gℓ=softmax(Wk·Fℓ)⊗Fℓ,
where Fℓ∈ℝC×H×W represents the axially aligned features of the *ℓ*-th layer in the current pyramid level. The learnable projections matrix Wk adaptively compresses the feature channels in a query-independent manner. After that, a 2D activation map is obtained by a normalization operation. We apply it to the original feature map and calculate the weighted average of all positions.

At this point, we have obtained the global context information of each layer. However, considering remote-sensing targets’ varying scale and complex context, the extracted global information still lacks inter-layer communication. Next, we use two parallel branches to enhance this global context information from the channel and hierarchical dimensions. The importance of each layer can be extracted as follow:(10)$${𝕎}_{level}=\sigma \left(W_{g}\overset{L}{\underset{\ell =1}{⋃}}G_{\ell }\right),$$
where ⋃ represents channel concatenation and $W_{g}$ is a learnable linear transformation matrices. Here, we consider the *L* groups of global contextual information as a whole to predict a vector ${𝕎}_{level}$, which represents the importance of each hierarchy. At the same time, we capture the inter-channel dependencies in another branch and apply the above vector to different layers, which are expressed as follows:(11)$$G^{{}^{\prime }}={𝕎}_{level}⊙\overset{L}{\underset{\ell =1}{⋃}}F_{s}\left(G_{\ell }\right),$$
where ⊙ represent element-wise product. The channel-wise feature selection $F_{s}\left(\cdot \right)$ first reduces the number of channels by using a 1 × 1 convolution followed by a layer normalization and ReLU activation function. In this way, important contextual information can be selectively retained, and useless channels are discarded so that the robustness of global contextual information is improved. Next, we use hierarchical weights ${𝕎}_{level}$ to re-weight channels of different layers, endowing the global contextual information with hierarchical-aware capability. Finally, we fuse feature maps from different layers with the scaled global context to obtain semantically aligned features:(12)$$Y=\sum_{\ell =1}^{L}\left(G_{\ell }^{{}^{\prime }}+F_{\ell }\right).$$

It is worth noting that the proposed alignment networks are based on neighbor features. Figure 5b visualizes the distribution of predicted quantities in the feature pyramid for different categories in the DOTA dataset. It can be found that each category concentrates on one and its adjacent level of the pyramid according to the target scale. Therefore, feature semantic alignment based on neighboring layers is the optimal level selection strategy to solve the semantic gap problem in the remote-sensing field. See Section 3.3.2 for detailed ablation experiments about selection strategies. On the one hand, our method only aligns 2× resampled neighbor features, which can significantly preserve the feature scale sensitivity. On the other hand, the weights of linear transformations (e.g., $W_{g}$ and $W_{k}$) in this module are shared within the hierarchy, so this co-optimization method can also alleviate the problem of semantic misalignment. More ablation experiments can be found in Section 3.3.5.

#### 2.2.4. Loss Function

The loss function of the network consists of three parts for different tasks: the classification loss for category prediction, the regression loss for oriented box localization, and the centerness loss for anchor quality evaluation. Compared with other mainstream alignment methods in the remote-sensing field, this network does not need to calculate the extra loss of subhead, so it is easier to optimize. The definition of the loss function is as follows:(13)$$L_{total}=\mu_{1}L_{cls}\left(p_{i},p_{i}^{*}\right)+\mu_{2}p_{i}^{*}L_{reg}\left(v_{i},v_{i}^{*}\right)+\mu_{2}p_{i}^{*}L_{ctr}\left(c_{i},c_{i}^{*}\right),$$
where the classification loss $L_{cls}$ is focal loss [6], $p_{i}$ represents the predicted category label at the i-th position of the feature map, $\mu_{1}$ and $\mu_{2}$ are the trade-off hyper-parameters and are both set to 1.0 by default, and $p_{i}^{*}$ is the point sample assigned by the central portion sampling strategy and is 1 if $p_{i}^{*}$ is a positive sample and 0 otherwise. The regression loss $L_{reg}$ is IoU loss [47]; $v_{i}$ represents the rotated box decoded from the predicted distance vector and angle, while $v_{i}^{*}$ is the ground truth box. The centerness loss $L_{ctr}$ is the cross entropy loss.

## 3. Results

### 3.1. Datasets

We conduct experiments on several remote-sensing datasets for oriented object detection, including DOTA1.0 [48], HRSC2016 [49], and UCAS-AOD [50]. Objects in all three datasets are labeled with oriented bounding boxes.

The DOTA dataset is a large-scale, publicly available image dataset for remote-sensing target detection with a total of 2806 images, ranging in size from 800 × 800 to 4000 × 4000. The dataset is annotated with 188,282 instances divided into 15 categories: bridge (BR), harbor (HA), ship (SH), plane (PL), helicopter (HC), small vehicle (SV), large vehicle (LV), baseball diamond (BD), ground track field (GTF), tennis court (TC), basketball court (BC), soccer field (SBF), roundabout (RA), swimming pool (SP), and storage tank (ST). The dataset is divided into three parts for training, validation, and testing with a ratio of 3:1:2, and the final test results are obtained through the official evaluation server. During training, we split the image into 1024 × 1024 sub-images with 200-pixel overlap.

HRSC2016 is an optical satellite dataset for oriented ship detection and includes 1061 images with size ranging from 300 × 300 to 1500 × 900. The images of this dataset were collected from six famous harbors that contain ships on the sea and close in-shore. The dataset is randomly divided into 436 images as the training set, 181 as the validation set, and 444 as the test set. In the experiments, we use the training set and validation set for training, and we evaluate the model on the test set in terms of PASCAL VOC07 metrics.

UCAS-AOD contains 1510 aerial images with two categories (car and plane) and 14,596 instances. The resolution of the images is 659 × 1280, with 510 images of cars and 1000 of airplanes. During training and inference, all images are resized to 800 × 1333 without changing the aspect ratios. For a fair comparison with the compared methods, we follow the same dataset splits setting in [51], where the ratio between the training set, validation set, and testing set is 5:2:3.

### 3.2. Implementation Details

We adopt the FPN-based ResNet50 [41] as our backbone network unless otherwise specified. The network is trained with a stochastic gradient descent (SGD) optimizer over a single GPU with two images per mini-batch. The weight decay and momentum are set to $10^{-4}$ and 0.9, respectively. The initial learning rate is set to 0.0025. Models are trained for 12 epochs, and the learning rate is reduced by a factor of 10 at the end of epochs 12, 16, and 19. We apply random horizontal flipping and random poly rotating with 0.5 probability for data augmentation. The inference time is calculated on a single GeForce RTX 3080 with 10GB memory. Our model is based on the MMRotate [52], an open-source toolbox for oriented object detection based on PyTorch.

### 3.3. Ablation Study

#### 3.3.1. Factor-by-Factor Experiment

In this section, we first give a macroscopic overview of the performance improvement and inference efficiency of the proposed spatial alignment, axial sampling, and semantic alignment compared to the baseline, as shown in Table 1. The structure and hyperparameters of each module are the optimal configurations obtained through more-detailed ablation experiments in the following sections. All ablation experiments are tested on the validation set of the DOTA dataset.

First, we used the naive center-based anchor-free detector as our baseline, which achieved 65.25% mAP and 23.8 FPS. After introducing the intra-level multi-scale spatial alignment module, mAP is improved by 2.43%. It can be seen that the network aligns the spatial location of features in different layers under the guidance of the leading feature. Subsequently, we introduced axially aware convolution for fine alignment. We achieved 0.21% mAP improvement and moderate inference speed, which shows that the proposed axially aligned sampling can cope with the varying orientation of remote-sensing targets to a certain extent, and the extracted features can characterize the rotated targets more effectively. Next, we add a hierarchical-wise semantic alignment network based on the spatially aligned features to verify the role of the semantic alignment module. The result shows a 0.6% mAP improvement with only a slight impact on inference speed, which verifies that this module can enhance intra-level semantics and alleviate cross-level semantic gaps. Finally, our OA-Det totally improves by 4.03% mAP while maintaining considerable efficiency compared to the baseline, solving the feature misalignment problem of the remote-sensing target from three dimensions. In subsequent experiments, we go deep inside each module to verify the effectiveness of the components.

#### 3.3.2. Evaluation of Spatial Alignment

For the proposed spatial alignment network, we tested various pyramid-level selection strategies to verify our point of view. The results are shown in Table 2.

We used the feature pyramid network as the baseline for this ablation experiment and further tested three different types of level selection strategies. The first strategy is to resample all pyramid features to the same resolution as $P_{4}$ and then perform feature refinement at all levels. This method samples up to 8x but only achieves a 0.94% improvement. Another strategy is to downsample high-resolution features to provide location information to upper features, which improves by 2.22% mAP compared to the baseline, validating the superiority of this bottom-up strategy. Next, our strategy, i.e., resampling the neighbor feature maps to the same resolution, achieves an impressive 2.3% mAP improvement. The result verifies that this strategy can provide semantic and location information to the leading level while maintaining the feature scale sensitivity. Finally, we test the effectiveness of introducing pyramid level $P_{2}$, which further improves 0.13% mAP with a negligible impact on inference speed. In summary, the proposed pyramid level selection strategy improves 2.43% mAP compared to the baseline with considerable speed. It verifies our point of view: spatial alignment of adjacent levels is optimal.

#### 3.3.3. Evaluation of Axially Aware Convolution

In this section, we verify the effectiveness of the proposed axially aware sampling method based on the multi-scale features extracted by the spatial alignment network. Five mainstream alignment methods are compared to test their axial alignment effects for remote-sensing targets with different orientations. Among them, regular convolution, grid sample, deformable convolution, and axially aware convolution are all sampling methods without additional detection heads. In addition, to compare with the alignment convolution, we designed an additional supervised version for the axially aware convolution, and the experimental results are shown in Table 3.

First, regular convolution achieves 66.67% mAP without feature aggregation of multi-scale features. Next, deformable convolution achieves an additional 0.34% mAP improvement over the regular grid sampling. Therefore, guiding sampling points can better alleviate the axial non-alignment problem of remote-sensing targets compared with directly predicting offset maps to interpolate features. However, the disadvantage of deformation convolution is the lack of offset restrictions, so it is easy to introduce a redundant background, as shown in Figure 7b. Then, we compare the performance of axially aware convolution and alignment convolution under sub-head guidance in the same network architecture. In this case, we aggregate features by a stack of convolution from the additional head instead of a 3 × 3 convolution. The results show that the axial convolution achieves a 0.21% mAP advantage. Although the accuracy decreases by 0.13% mAP compared with the unsupervised version of axially aware convolution, it still outperforms the alignment convolution, which shows the effectiveness of the proposed axial alignment method. After introducing the spatially aware block, the performance exceeds that of the supervised version by 0.2% mAP, proving that the large kernel can provide broader spatial information for axial prediction tasks. The visualization results are shown in Figure 7d. Axially aware convolution makes up for the shortcomings of alignment convolution, which can detect more small objects and align their orientations without introducing explicit supervision signals and detection heads.

Next, we evaluate the mean Average Orientation Error (mAOE) of the above methods, validating their axial alignment ability for different categories in DOTA. As shown in Table 4, our axially aware convolution outperforms other compared feature alignment methods and achieves minimal orientation errors on BD, GTF, LV, SP, and HC, demonstrating excellent axial alignment performance for remote-sensing targets with high aspect ratios and varying shapes.

#### 3.3.4. Evaluation of Collaborative Optimization Strategy

Our axially aware convolution achieves better performance without explicit signal guidance due to the shared weight co-optimization strategy to improve the robustness of the predicted axis. This section conducts comparative experiments on different weight-sharing strategies of the spatial alignment network. The results are shown in Table 5.

First, we test the effect of the collaborative optimization strategy on spatial alignment. The results show that after sharing the offset weight, the performance achieved a significant improvement of 1.23%; so it can be proven that sharing offsets for adjacent layer features can help spatial alignment. Next, we focus on the axis-aware convolution of the predicted axis and sampling weight. For the sampling point weights, we find that the co-optimization strategy can achieve a slight accuracy improvement. Since most axially aligned sampling points are highly context-related compared to sparse sampling, sharing mask weight can make full use of the multi-scale features within the layer to distinguish foreground from background, thus further downweighting the sampling points still in the background. When optimizing the axial prediction task separately, the overall performance decreases. However, when axial weights are shared across levels, the network outperforms the previous stage by 0.41% mAP, proving that this strategy of sharing weights across layers can make axial prediction more robust. So far, we have verified the validity of all components inside the spatial alignment network.

#### 3.3.5. Evaluation of Semantic Alignment

In this section, we perform ablation experiments on the proposed semantic alignment networks to verify the effectiveness of each internal component one-by-one, and the results are shown in Table 6. For a fair comparison, we first construct the network using a naive approach instead of the proposed components, where global average pooling (GAP) is performed to the extracted global context, and a 1 × 1 convolution is used to perform simple linear projection, achieving 65.13% mAP.

Next, we use the proposed query-independent global context modeling to replace GAP. We found that a 2.4% mAP improvement is achieved with almost no parameter increase. We further introduce channel-wise feature selection, which outperforms the naive method while reducing the parameter by 0.13M. Finally, the introduction of hierarchical-wise reweighting improves the accuracy by 0.71%, proving that this component empowers global context hierarchy awareness. Compared with the network of the naive method, our module brings an overall improvement of 3.15% mAP. At this point, we have verified the rationality of the internal design of our semantic alignment network, which can alleviate the semantic gap problem and strengthen the exchange of information between levels.

### 3.4. Main Results and Analysis

#### 3.4.1. Results on DOTA

We show the performance of state-of-the-art methods on the DOTA dataset for two-stage, single-stage, and anchor-free detectors in Table 7. Our method achieves excellent performance of 74.82% and 78.11% mAP in single-scale and multi-scale tests, respectively, with ResNet101 as the backbone.

Note that our OA-Det achieves the best performance among the compared methods in the categories of small vehicle (SV), large vehicle (LV), and ship (SH), which shows that our method is good at dealing with small targets with different orientations in dense scenarios. Even more surprising is that we have a huge lead in the harbor (HA) and swimming pool (SP), which demonstrates that for objects with different shapes in complex backgrounds, our method can better perceive their orientations to obtain accurate regression. The visualization results of the DOTA dataset are shown in Figure 8.

#### 3.4.2. Results on HRSC2016

We validate the effectiveness of our OA-Det on the HRSC2016 dataset. The slender ships in the dataset are challenging for anchor-free detectors, but we still achieve excellent performance with the feature alignment network.

As shown in Table 8, the results show that our OA-Det outperforms the compared anchor-free methods (e.g., RepPoints and CenterNet) and even anchor-based methods by 90.10% mAP. The visualization results of the HRSC2016 dataset are shown in Figure 9. With the proposed alignment network, we can accurately regress high-aspect-ratio objects and correctly distinguish ships from harbors.

#### 3.4.3. Results on UCAS-AOD

The UCAS-AOD dataset contains objects with different orientations and scales. Table 9 shows that our OA-Det achieves the best performance among the compared methods. Note that Yolov3 [69] and Faster-RCNN [4] are methods based on horizontal bounding boxes, so we add angle prediction to the above two methods. The results show that our OA-Det achieves state-of-the-art performance of 90.29% mAP, outperforming other compared one-stage and two-stage detectors in both categories. The visualization of the dataset is shown in Figure 10.

## 4. Discussion

After analyzing the above ablation experiment and the main results from shallow-to-deep, we verified the effectiveness of each module and its internal components from several aspects. Nevertheless, we still find that the proposed method needs to be improved in some categories. For targets with extreme aspect ratios (e.g., harbors and bridges), although our alignment network can sense the target’s orientation, anchor-free detectors usually cannot accurately return the entire target. In the future, we will do further in-depth research on the OBB representation of anchor-free detectors. Further, transformer-based algorithms have achieved excellent performance in many arbitrarily oriented object detection networks. We will also try to use the Swin transformer [1] to solve the problems in the remote-sensing field.

## 5. Conclusions

Aiming at the problem of multi-scale feature misalignment in oriented object detection, this paper analyzes the root causes of the problem from three dimensions and gives the corresponding efficient solutions. First, we propose an intra-level collaborative feature alignment network that can absorb the semantic knowledge and localization information under different receptive fields by sparse sampling and axially aware convolution. In this way, the orientations of instances and spatial information of pyramid features can be aligned in a coarse-to-fine manner without additional supervised signals. Based on this, we further propose a hierarchical-wise global context module that alleviates the semantic gap of multi-scale features from the channel- and pyramid-level dimensions. Detailed ablation experiments and excellent performance on three testbeds demonstrate the effectiveness of the proposed method.

## Figures and Tables

**Figure 1 sensors-23-00207-f001:**
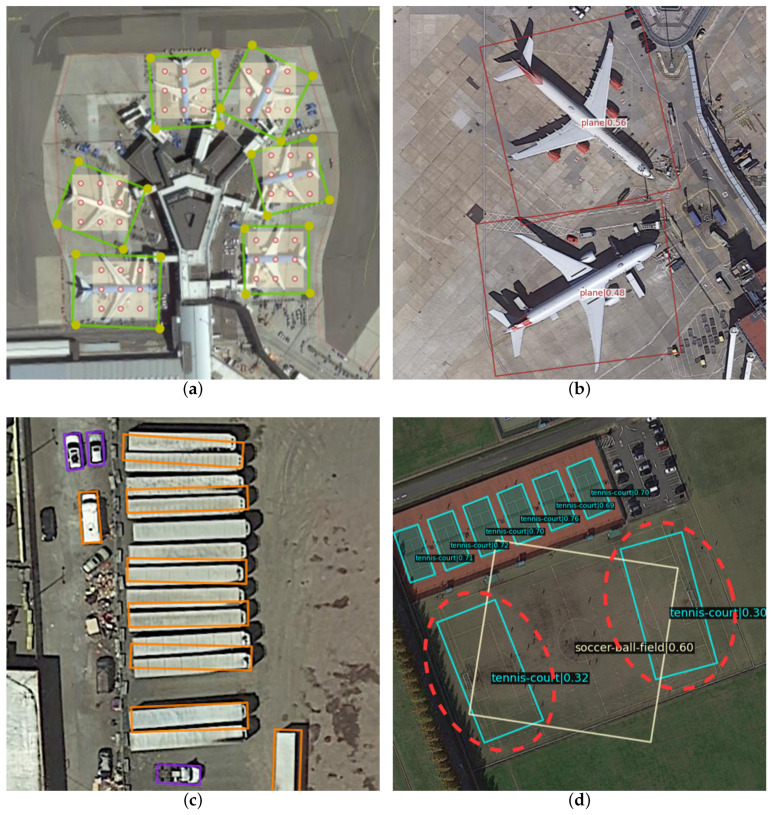
(**a**) The sampling axis of the regular convolution does not match the remote-sensing target with different orientations. The green box is the ground truth, the yellow points are the four vertices, and the red points are the sampling points of the convolution. (**b**) Axial misalignment leads to regression error in predicting rotated boxes. (**c**) Missed detection of densely arranged remote-sensing targets due to spatial misalignment. (**d**) Classification errors caused by semantic misalignment(red dashed circle). A portion of the soccer field was misdetected as a tennis court.

**Figure 2 sensors-23-00207-f002:**
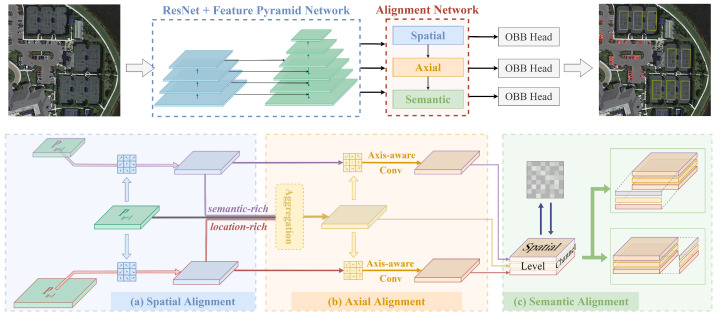
The overall architecture of our OA-Det for oriented object detection consists of a backbone network, a feature alignment network, and an anchor-free head network. The ResNet-based feature pyramid network extracts common features from an RGB aerial image. The feature alignment network sequentially aligns pyramid features from three dimensions: (**a**) an intra-level collaborative alignment network in the spatial dimension, (**b**) an axis-aware convolution in the axial dimension, and (**c**) a hierarchical-wise context fusion module in the semantic dimension. Finally, aligned features are sent to the classification head and the regression head to obtain the OBB prediction result.

**Figure 3 sensors-23-00207-f003:**
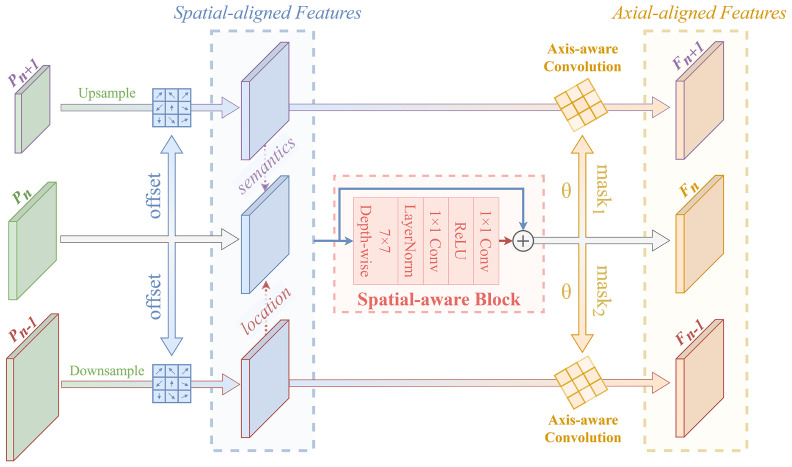
Structure of intra-level collaborative feature alignment network.

**Figure 4 sensors-23-00207-f004:**
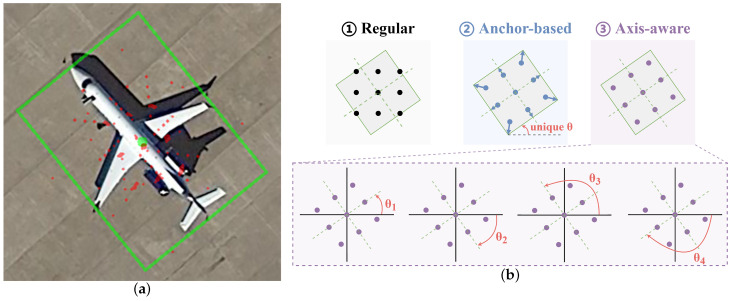
(**a**) The green box is the ground truth, and the green point is its center point. The red point is the sampling position guided by the anchor on the pyramid features from P3 to P6 based on the green point. It can be seen that most of the sample points are offset to the background under the guidance of the anchor box. (**b**) Sampling axis representation under three sampling methods. The regular sampling method cannot align the axis of remote-sensing targets, and the anchor-aware sampling axis is represented by a unique angle. Our axially aware sampling can represent an axis in four ways.

**Figure 5 sensors-23-00207-f005:**
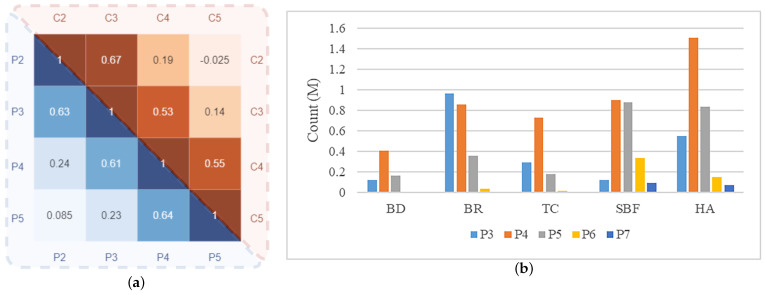
(**a**) Correlation matrix of different layers. The blue in the lower left corner and the red in the upper right corner are the correlations of the backbone features and pyramid features, respectively. (**b**) Pyramid-level distribution of different categories predicted by baseline. The colors of the columns represent different pyramid levels, and the categories include baseball diamond (BD), bridge (BR), tennis count (TC), soccer field (SBF), and harbors (HA).

**Figure 6 sensors-23-00207-f006:**
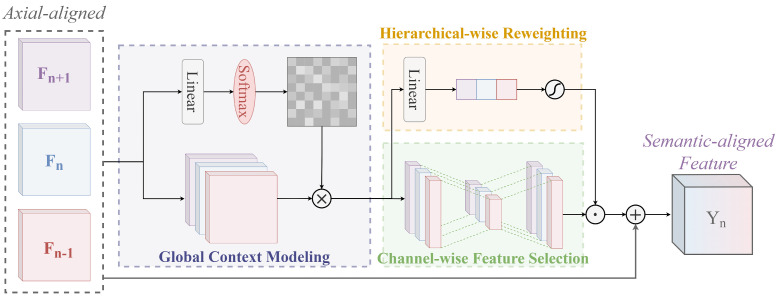
The structure of hierarchical-wise context fusion module.

**Figure 7 sensors-23-00207-f007:**
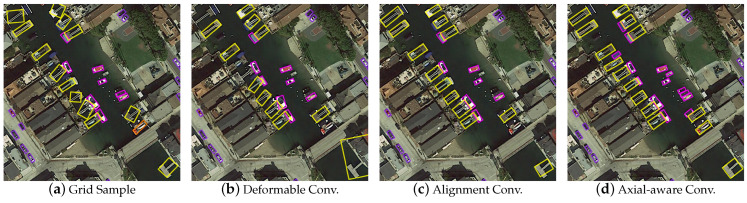
Visualization of different feature alignment methods.

**Figure 8 sensors-23-00207-f008:**
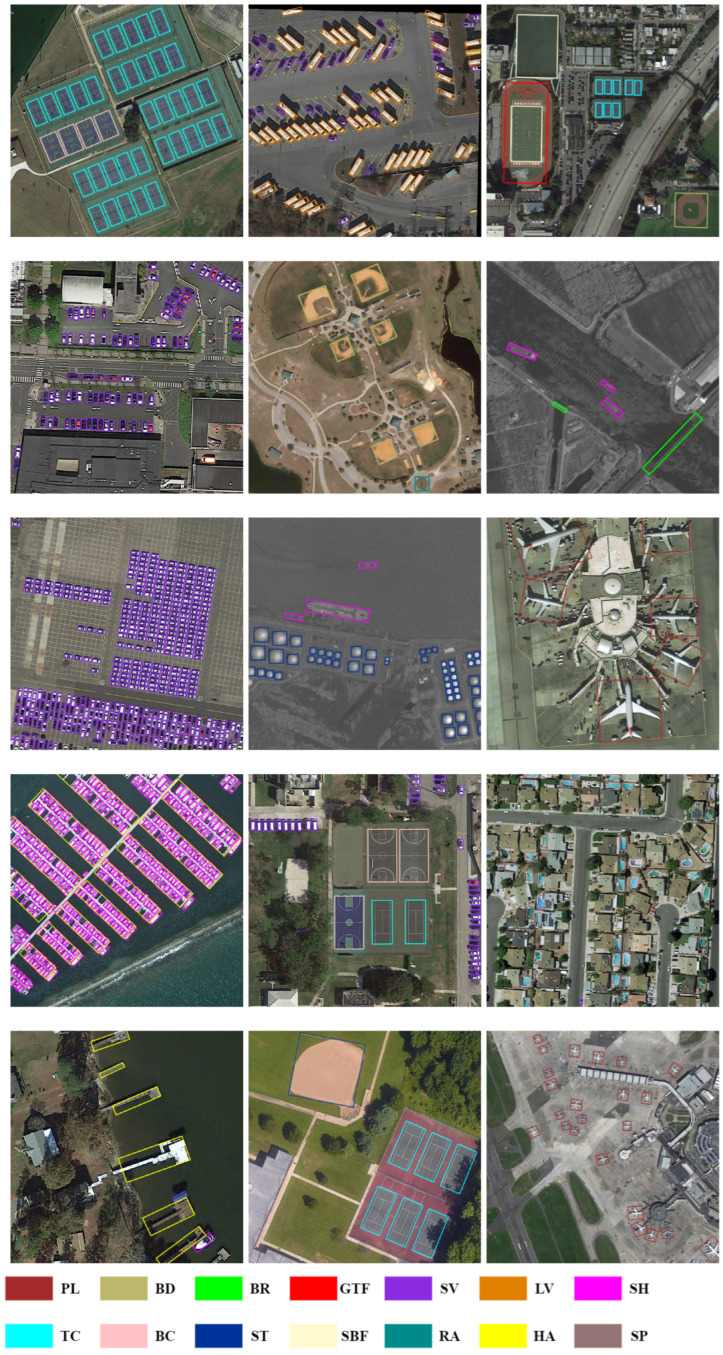
Visualization of predictions on the DOTA dataset using our method OA-Det.

**Figure 9 sensors-23-00207-f009:**
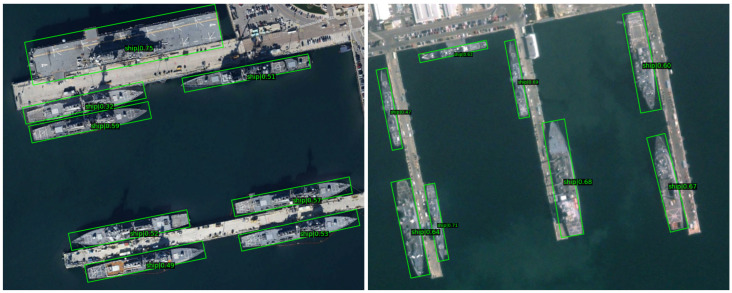
Visualization of predictions on the HRSC2016 dataset using our OA-Det method.

**Figure 10 sensors-23-00207-f010:**
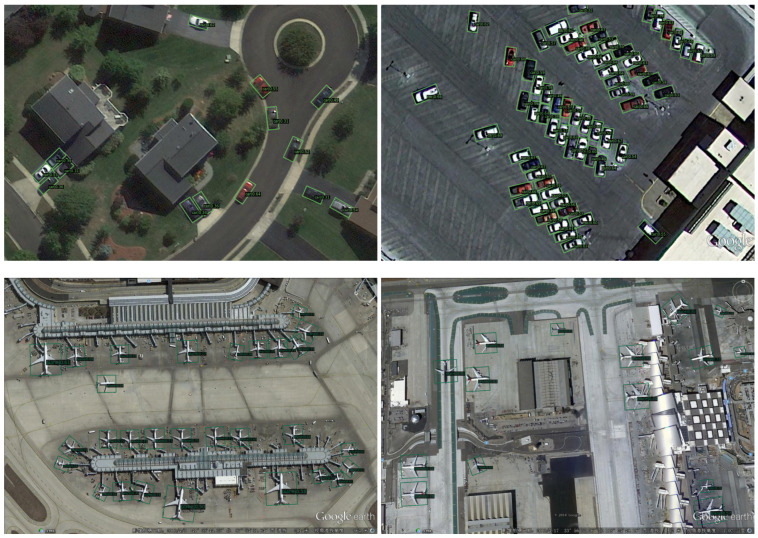
Visualization of predictions on the UCAS-AOD dataset using our OA-Det.

**Table 1 sensors-23-00207-t001:** Evaluation of different components: ‘✓’ indicates that the component is added on the basis of the baseline, and ‘✕’ indicates that the component is not enabled.

	Baseline	Different Settings of OA-Det			
+ Spatial Alignment	✕	✓	✓	✓	✓
+ Axial Sampling	✕	✕	✓	✕	✓
+ Semantic Alignment	✕	✕	✕	✓	✓
**mAP(%)**	65.25	67.68	67.89	68.28	69.28
**FPS**	23.8	21.6	15.0	19.7	14.3

**Table 2 sensors-23-00207-t002:** Multi-scale hierarchical selection strategy within different pyramid features.

The Strategy of Alignment Layer	Max Resize Factor	Number of Layers	mAP(%)	FPS
baseline	-	-	65.25	23.8
all level	8x	5	66.19	23.2
lower level	2x	2	67.47	21.9
neighbor level	2x	3	67.55	20.9
neighbor level with P2	2x	3	67.68	20.5

**Table 3 sensors-23-00207-t003:** Performance comparison of different feature alignment methods.

Alignment Methods	Additional Sub Head	Feature Aggregation	mAP(%)
Regular Conv.	✕	-	66.67
Grid Sample	✕	3 × 3 conv	67.00
Deformable Conv.	✕	3 × 3 conv	67.34
Alignment Conv.	✓	stack of conv	67.48
Axially aware Conv.	✓	stack of conv	67.69
Axially aware Conv.	✕	3 × 3 conv	67.56
Axially aware Conv.	✕	spatially aware block	67.89

**Table 4 sensors-23-00207-t004:** Ablation experiment of average orientation error of mainstream feature alignment methods on DOTA1.0 dataset. All results are tested on the validation set.

Methods	PL	BD	BR	GTF	SV	LV	SH	TC	BC	ST	SBF	RA	HA	SP	HC	mAOE
Regular Conv	19.90	14.96	6.20	2.82	6.45	4.90	6.06	2.78	6.85	6.20	4.18	9.07	5.37	10.72	20.34	8.45
Grid Sample	17.54	17.08	5.25	7.81	5.66	4.60	6.00	2.18	4.97	4.93	2.99	10.54	5.68	8.17	15.48	7.93
DeformConv	16.83	18.23	4.02	5.44	5.58	4.01	5.47	1.57	6.81	5.28	3.67	7.37	6.51	7.57	18.30	7.78
AlignConv	14.68	16.83	3.62	2.48	4.74	3.00	4.63	3.02	3.96	5.86	5.92	7.88	4.42	8.27	22.03	7.42
Axially aware Conv	16.88	12.95	4.67	2.33	5.61	2.98	4.78	3.93	4.25	5.90	3.97	8.47	5.02	6.85	13.08	6.78

**Table 5 sensors-23-00207-t005:** Effectiveness of shared weights.

Offset		Mask		Angle		mAP(%)
Separate	Shared	Separate	Shared	Separate	Shared	
✓						66.25
	✓					67.48
	✓	✓		✓		66.35
	✓		✓	✓		66.41
	✓		✓		intra-level	67.79
	✓		✓		cross-level	67.89

**Table 6 sensors-23-00207-t006:** Effectiveness of internal components of semantic alignment network.

Global Context	Channel-Wise	Hierarchical-Wise	mAP(%)	Param(M)
avg pooling	single conv	✕	65.13	42.53
✓	single conv	✕	67.53	42.53
✓	✓	✕	67.57	42.40
✓	✓	✓	68.28	44.78

**Table 7 sensors-23-00207-t007:** Comparisons with other state-of-the-art methods on DOTA dataset. The data in each column indicate the average precision of each category in percent. The last column indicates the mean-Average-Precision of the method: ‘†’ represents the results for multi-scale training and testing. The results with red color denote the best results in each column.

Methods	Backbone	PL	BD	BR	GTF	SV	LV	SH	TC	BC	ST	SBF	RA	HA	SP	HC	mAP
*Two-stage*																	
FR-O [48]	R-101	79.42	77.13	17.70	64.05	35.30	38.02	37.16	89.41	69.64	59.28	50.30	52.91	47.89	47.40	46.30	54.13
ICN [53]	R-101	81.40	74.30	47.70	70.30	64.90	67.80	70.00	90.80	79.10	78.20	53.60	62.90	67.00	64.20	50.20	68.20
RoI-Trans. [54]	R-101	88.64	78.52	43.44	75.92	68.81	73.68	83.59	90.74	77.27	81.46	58.39	53.54	62.83	58.93	47.67	69.56
SCRDet [27]	R-101	89.98	80.65	52.09	68.36	68.36	60.32	72.41	90.85	** 87.94 **	** 86.86 **	65.02	66.68	66.25	68.24	65.21	72.61
FADet [55]	R-101	** 90.21 **	79.58	45.49	76.41	73.18	68.27	79.56	90.83	83.40	84.68	53.40	65.42	74.17	69.69	64.86	73.28
Gliding Vertex [56]	R-101	89.64	85.00	52.26	** 77.34 **	73.01	73.14	86.82	90.74	79.02	86.81	59.55	** 70.91 **	72.94	70.86	57.32	75.02
Mask OBB [57]	RX-101	89.56	** 85.95 **	54.21	72.90	76.52	74.16	85.63	89.85	83.81	86.48	54.89	69.64	73.94	69.06	63.32	75.33
CenterMap-O [58]	R-101	89.83	84.41	** 54.60 **	70.25	77.66	78.32	87.19	90.66	84.89	85.27	56.46	69.23	74.13	71.56	66.06	76.03
*Single-stage*																	
PIoU [59]	DLA-34	80.9	69.7	24.1	60.2	38.3	64.4	64.8	90.9	77.2	70.4	46.5	37.1	57.1	61.9	64.0	60.5
A2S-Det [60]	R-101	89.59	77.89	46.37	56.47	75.86	74.83	86.07	90.58	81.09	83.71	50.21	60.94	65.29	69.77	50.93	70.64
DAL [61]	R-101	88.61	79.69	46.27	70.37	65.89	76.10	78.53	90.84	79.98	78.41	58.71	62.02	69.23	71.32	60.65	71.78
CFC-Net [28]	R-50	89.08	80.41	52.41	70.02	76.28	78.11	87.21	** 90.89 **	84.47	85.64	60.51	61.52	67.82	68.02	50.09	73.50
R${}^{3}$Det [13]	R-101	88.76	83.09	50.91	67.27	76.23	80.39	86.72	90.78	84.68	83.24	61.98	61.35	66.91	70.63	53.94	73.79
SLA [62]	R-50	88.33	84.67	48.78	73.34	77.47	77.82	86.53	90.72	86.98	86.43	58.86	68.27	74.10	73.09	** 69.30 **	76.36
*Anchor-free*																	
CenterNet-O [24]	H-101	89.02	69.71	37.62	63.42	65.23	63.74	77.28	90.51	79.24	77.93	44.83	54.64	55.93	61.11	45.71	65.04
P-RSDet [63]	R-101	89.02	73.65	47.33	72.03	70.58	73.71	72.76	90.82	80.12	81.32	59.45	57.87	60.79	65.21	52.59	69.82
O2-DNet [64]	H-104	89.31	82.14	47.33	61.21	71.32	74.03	78.62	90.76	82.23	81.36	60.93	60.17	58.21	66.98	61.03	71.04
BBAVector [65]	R-101	88.35	79.96	50.69	62.18	78.43	78.98	87.94	90.85	83.58	84.35	54.13	60.24	65.22	64.28	55.70	72.32
DRN [66]	H-104	89.71	82.34	47.22	64.10	76.22	74.43	85.84	90.57	86.18	84.89	57.65	61.93	69.30	69.63	58.48	73.23
AOPG [10]	R-101	89.14	82.74	51.87	69.28	77.65	82.42	88.08	90.89	86.26	85.13	60.60	66.30	74.05	67.76	58.77	75.39
*Ours*																	
OA-Det	R-101	89.51	76.38	51.47	68.38	73.37	77.05	87.85	90.69	82.38	86.23	63.09	65.06	75.68	76.22	58.92	74.82
OA-Det${}^{†}$	R-101	88.82	84.02	53.49	70.42	** 81.42 **	** 84.27 **	** 88.67 **	90.71	83.90	86.33	** 66.74 **	69.79	** 77.73 **	** 80.65 **	64.66	** 78.11 **

**Table 8 sensors-23-00207-t008:** Comparisons with high-quality detection performance on HRSC2016 dataset: **mAP${}_{07}$** indicates the results under VOC2007 mAP metrics.

Methods	Backbone	Image Size	mAP${}_{07}$(%)
R${}^{2}$CNN [25]	ResNet101	800 × 800	73.07
Axis Learning [67]	ResNet101	800 × 800	78.15
RRD [25]	VGG16	800 × 800	84.30
RepPoints [25]	ResNet101	512 × 800	85.16
RoI-Transformer [25]	ResNet101	512 × 800	86.20
CenterNet-O [25]	DLA-34	800 × 800	87.89
Gliding Vertex [56]	ResNet101	512 × 800	88.20
CSL [68]	ResNet101	800 × 800	89.62
OA-Det(Ours)	ResNet101	800 × 800	90.10

**Table 9 sensors-23-00207-t009:** Comparisons with high-quality detection performance on UCAS-AOD dataset.

Methods	Backbone	Car	Plane	mAP(%)
Yolov3 [69]	Darknet53	74.63	89.52	82.08
RepPoints [25]	ResNet101	83.02	89.34	86.18
Faster-RCNN [4]	ResNet50	86.87	89.86	88.36
RoI-Transformer [54]	ResNet101	87.99	89.90	88.95
CFC-Net [28]	ResNet101	89.29	88.69	89.49
DAL [61]	ResNet101	89.25	90.49	89.87
S${}^{2}$A-Net [11]	ResNet50	89.56	90.42	89.99
OA-Det(Ours)	ResNet101	90.02	90.56	90.29

## Data Availability

The source code of the paper is available at https://github.com/Virusxxxxxxx/OA-Det, accessed on 20 November 2022.

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
