# Peer review of "Shared-Weight-Based Multi-Dimensional Feature Alignment Network for Oriented Object Detection in Remote Sensing Imagery"

_sensors, 2022, doi:10.3390/s23010207_

Round 1

Reviewer 1 Report

The manuscript is well-written and well-structured. The authors focus on solving the problems of feature misalignment in the spatial, axial and semantic dimensions, respectively. The ideas presented are reasonable and the results are clear and convincing. I only have a few comments as follows:

i. The figures are too blur and too low resolution to recognize the text inside.

ii In line 271, “where L represents the number of spatial-aligned features within the current pyramid level.” However, according to function (4), “l” denotes the pyramid level, not the feature. Is the definition of “L” incorrect?

iii. In line 276, What does the MLP stand for?

iv. In function (8), “delta P(n) = …” should be "delta P(n)k = …” If I am not mistaken.

v. In table. 5, the addition of the shared angle weight has obvious benefit, but that of the mask weight does not seem to help. I would like to know if the authors have any further experiments or explanations on this.

Reviewer 2 Report

1, The resolution of Figures used in the paper should be further improved.

2, Abbreviations (e.g., FPN) should give their full names when they are first motioned.

3, Pls give detailed descriptions of feature pyramid network about its architecture and functions.

4, One of the main contributions of this paper is designing a shared-weight network, pls explain the advantage of share-weight strategy and how to remedy the accuracy loss of this strategy.

5, Figure 5, wrong format of the caption.

6, Pls explain whether your learning dataset is orientation corrected, if not, how to deal with the orientation errors?

6, In equation (13), three types of loss are made up together, why are the balance factors of these loss not considered?  Since different types of loss have different units.

7, Pls give the websites of the resources of the learning datasets.

8, Table 1, unclear meanings of the symbols '√' and '×'.

9, Table 7, please give a special mark on the best result. Apart from the last column (i.e., mPA), numbers in other columns are ambiguous. Pls indicate that the numbers are percentages, and do the numbers represent precision? recall? 

10. Table 8 and 9, what is the meaning of mAP07?

11. Would you like to make your code publicly available? Research works in computer vision have the convention of 'open resource'.
